# Management of High-Risk Neuroblastoma with Soft-Tissue-Only Disease in the Era of Anti-GD2 Immunotherapy

**DOI:** 10.3390/cancers16091735

**Published:** 2024-04-29

**Authors:** Maite Gorostegui, Juan Pablo Muñoz, Sara Perez-Jaume, Margarida Simao-Rafael, Cristina Larrosa, Moira Garraus, Noelia Salvador, Cinzia Lavarino, Lucas Krauel, Salvador Mañe, Alicia Castañeda, Jaume Mora

**Affiliations:** Pediatric Cancer Center Barcelona, Hospital Sant Joan de Déu, 08950 Barcelona, Spain; maite.gorostegui@sjd.es (M.G.); juanpablo.munoz@sjd.es (J.P.M.); margarida.simaorafael@sjd.es (M.S.-R.); cristina.larrosa@sjd.es (C.L.); moira.garraus@sjd.es (M.G.); noelia.salvador@sjd.es (N.S.); cinzia.lavarino@sjd.es (C.L.); lucas.krauel@sjd.es (L.K.); smane@quironsalud.es (S.M.); alicia.castanedah@sjd.es (A.C.)

**Keywords:** neuroblastoma, locoregional disease, INRG, INSS, anti-GD2 immunotherapy, first complete remission, recurrence, genetic features

## Abstract

**Simple Summary:**

Neuroblastoma (NB) presents with two patterns of disease: with or without metastasis. Both types of disease presentation include tumors with high-risk (HR) features. The management of HR-NB includes chemotherapy, surgery, radiotherapy, and anti-GD2 immunotherapy. Anti-GD2 monoclonal antibodies (mAbs) have significantly improved the outcome of HR-NB patients but they are mostly effective against disease affecting the bone and/or bone marrow (known as the osteomedullary compartment), and less so against soft tissue disease. The question arises as to whether anti-GD2 immunotherapy might benefit HR-NB patients with disease compounded by only soft tissue. In this retrospective review, we found that achieving first complete remission with chemotherapy, surgery, and radiotherapy does not prevent the risk of relapse. However, adding anti-GD2 mAbs once the patient has achieved complete remission significantly decreases the chances of relapse by 80%. Our study provides further support to indicate anti-GD2 mAbs in all cases with HR-NB.

**Abstract:**

Neuroblastoma presents with two patterns of disease: locoregional or systemic. The poor prognostic risk factors of locoregional neuroblastoma (LR-NB) include age, *MYCN* or *MDM2-CDK4* amplification, 11q, histology, diploidy with *ALK* or *TERT* mutations, and *ATRX* aberrations. Anti-GD2 immunotherapy has significantly improved the outcome of high-risk (HR) NB and is mostly effective against osteomedullary minimal residual disease (MRD), but less so against soft tissue disease. The question is whether adding anti-GD2 monoclonal antibodies (mAbs) benefits patients with HR-NB compounded by only soft tissue. We reviewed 31 patients treated at SJD for HR-NB with no osteomedullary involvement at diagnosis. All tumors had molecular genetic features of HR-NB. The outcome after first-line treatment showed 25 (80.6%) patients achieving CR. Thirteen patients remain in continued CR, median follow-up 3.9 years. We analyzed whether adding anti-GD2 immunotherapy to first-line treatment had any prognostic significance. The EFS analysis using Cox models showed a HR of 0.20, *p* = 0.0054, and an 80% decrease in the risk of relapse in patients treated with anti-GD2 immunotherapy in the first line. Neither EFS nor OS were significantly different by CR status after first-line treatment. In conclusion, adding treatment with anti-GD2 mAbs at the stage of MRD helps prevent relapse that unequivocally portends poor survival.

## 1. Introduction

Neuroblastoma (NB) is a pediatric cancer that arises from precursor cells of the peripheral sympathetic nervous system. Neuroblastoma is clinically heterogeneous and may present with two well recognized patterns of disease: (A) locoregional (International Neuroblastoma Staging System [INSS] Stages 1, 2, and 3; and International Neuroblastoma Risk Group (INRG) classification system L1 and L2); and (B) systemic (INSS stage 4/4S or INRG stage M/MS). Both disease presentations include low- and high-risk subgroups defined by clinical-biological variables that are relatively well defined within the major cooperative groups [1,2].

Locoregional neuroblastoma (LR-NB) is characterized by lack of distant metastases and represents approximately 40% of NB at presentation. Most commonly, patients present asymptomatic or with symptoms related to local compression of the tumor mass to neighboring organs. These tumors usually have a good prognosis even without cytotoxic therapy. However, some will recur locally, or worse yet, progress to stage 4/M disease [3]. The risk factors that predict disease progression or recurrence in LR-NB known to date include age, MYCN amplification, 11q status, histology, and DNA ploidy [1]. A retrospective analysis of the MSKCC cohort of conservatively managed LR-NB identified that MYCN amplification strongly associated with diploid DNA index, the best predictor of adverse clinical outcome in this cohort [4]. As such, ploidy is now part of the stratification criteria used by the COG [5]. More recently, aside from amplified-*MYCN*, *ALK* mutations, *ATRX* aberrations, *MDM2-CDK4* amplification, and *TERT* mutations have been associated with adverse prognosis and thus with HR disease [6,7,8,9]. Despite this, the excellent survival rate of LR-NB patients suggests that LR-NB identified according to the current clinical staging by INRG is strongly associated with low-risk biology and therefore can be successfully managed without cytotoxic therapy [1].

Systemic or metastatic disease (so called INSS stage 4 or INRG stage M) represents approximately 60% of all NB. This phenotype is biologically defined by tumor cells with the ability to metastasize distally. The ability of the tumor cells to reach certain organs correlates with clinical outcomes [10]. For instance, the involvement of cortical bone and/or bone marrow (BM) is one of the major definitions of high-risk disease, occurring respectively in >60% and 80% of stage 4/M cases [11]. However, a minor subset of stage 4/M cases will involve distant metastases limited to lymph nodes (LNs), the so-called stage 4N (nodal), usually with a more indolent course [12]. NB metastasizes to bone (B)/BM hematogenously, but nodal invasion occurs via the lymph node drainage chain or the thoracic duct. The latter accounts for the predominance of left neck/supraclavicular sites in 4N cases [12]. The pattern of metastasis is also well circumscribed in another subgroup of disseminated cases, the stage 4S/MS (S as Special). Stage 4S/MS cases are characterized by a metastatic pattern limited to skin, liver, and BM without bone involvement [10]. Most recently, a dissemination pattern involving the skeletal muscle has been described and found to resemble the 4S/MS phenotype with similar low-risk biological characteristics and favorable prognosis [13].

Whether locoregional or systemic, approximately 50% of NB present with high-risk (HR) disease [14,15,16]. High-risk disease is most frequently associated with age (older than 18 months) and the pattern of metastases in the osteomedullary compartment, B/BM [17,18,19,20,21]. Accordingly, the unique pattern of distant metastasis for the small subset of patients with stage 4N, with no B/BM involvement, has been correlated with superior event-free survival (EFS). In fact, 4N patients stand out as virtually the only survivors of metastatic HR-NB in the era before myeloablative therapy (MAT) and anti-GD2 immunotherapy became standard therapies [12,22]. A study of stage 4 patients who received MAT but no anti-GD2 immunotherapy showed a better prognosis among those with only extra-skeletal disease and no B/BM metastases [23]. A large retrospective study from the INRG found the 4N subgroup to have significantly better EFS and overall survival (OS) compared to other stage 4/M patients even though the 4N patients received less intensive treatment [12]. The INRG investigators recommended considering less intensive therapy for 4N but the most recent risk classification system did not clear 4N [5]. Genetic aberrations in this small subgroup of stage 4 patients seem not to be different from osteomedullary-involved stage 4/M disease. In a recent study, the frequencies of *ALK* and *ATRX* defects were as expected for HR-NB. The frequency of *MDM2-CDK4* co-amplification seemed to be greater and *TERT* aberrations were the most common aberration after *MYCN* amplification [8].

Gangliosides are carbohydrate-containing sphingolipids (glycosphingolipids) embedded in the outer cell membrane, carrying on the outside two or three monosaccharide units terminating in one or two N-acetylneuraminic acids (Neu5Ac or NANA) [24]. The ganglioside GD2 carries two NANA (disialoganglioside) that participate in the interaction with membrane proteins and lipids to regulate cellular signaling [25], while facilitating cell–cell recognition and adhesion [26]. Disialoganglioside 2 (GD2) is expressed in neural and mesenchymal stem cells during fetal development. Postnatal expression in healthy tissues is restricted to the peripheral neurons, central nervous system, and skin melanocytes [27]. The density of GD2 in neuroblastoma is unusually high, with millions of molecules per cell [28]. Lower GD2 expression can also be found in osteosarcoma [29,30,31,32], melanoma, and some brain tumors [33,34,35]. 

Anti-GD2 monoclonal antibodies (mAbs) bind GD2-expressing tumor cells, engage FcR-bearing myeloid effectors to perform antibody-dependent cell-mediated phagocytosis, engage FcR-bearing natural killer cells to perform Ab-dependent cell-mediated cytotoxicity, activate complements to perform complement-dependent cytotoxicity, and may even cause direct induction of apoptosis [36]. The first-in-human anti-GD2 mAb, the murine 3F8, was developed in 1985 [37,38] and published in 1987 [39]. After demonstrating its potential against BM disease in patients with primary refractory NB, or those in complete remission [40], mouse 3F8 was humanized [36], brought to the clinic in 2011, and FDA-approved in 2020 with the name naxitamab (Danyelza) [41].

Anti-GD2 immunotherapy has significantly improved the outcome of HR-NB patients and is mostly effective against osteomedullary disease, but less so against soft tissue disease [40,42]. Current standard therapy for HR-NB comprises a backbone of induction chemotherapy and surgery, followed by consolidation with MAT and autologous stem cell transplant (ASCT), and post-consolidation therapy with radiotherapy and anti-GD2-based immunotherapy. The relevance of each of these components has been dissected recently, and the role of MAT and ASCT in the era of anti-GD2 immunotherapy is, at least, questionable since no randomized trial has demonstrated an impact in OS. Clinical trials using anti-GD2 immunotherapy demonstrated a significant improvement in EFS and OS in the context of minimal residual disease (MRD) [43,44]. Most recently, naxitamab has demonstrated efficacy in eradicating clinically measurable disease when limited to B and/or BM [45]; however, its efficacy remains limited against soft tissue disease. Only recently has chemo-refractory soft tissue NB been shown to be responsive to anti-GD2 immunotherapy when combined with chemotherapy [46]. Indeed, recent strategies of chemo-immunotherapy have been adopted to overcome early resistance in HR-NB [47,48]. 

While anti-GD2 mAbs alone are effective against B/BM disease and the benefit regarding soft tissue disease is less certain, the question arises as to whether anti-GD2 immunotherapy might benefit high-risk LR-NB and stage 4N patients given that the disease is compounded by only soft tissue. In this study, we retrospectively reviewed all patients with HR-NB referred to our center for treatment with anti-GD2 immunotherapy diagnosed with non-osteomedullary subtypes of disease. Our goal was to ascertain the relevance of each of the treatment modalities included in the management of HR-NB patients with disease affecting only the soft tissue compartment.

## 2. Patients and Methods

This retrospective report covers all patients from June 2017 until December 2023, treated at SJD Barcelona Children’s Hospital for HR-NB with no osteomedullary involvement at diagnosis, i.e., LR-NB or stage 4N (stage 4/M NB based solely on distant LN involvement and without metastases in B/BM). High-risk disease was defined as follows: *MYCN* non-amplified stage 4N diagnosed at age ≥18 months or *MYCN*-amplified stage 4N at any age; *MYCN*-amplified LR-NB at any age or *MYCN* non-amplified LR-NB diagnosed at age ≥18 months and with biological HR features (diploidy plus any of the following: segmental copy number variations (CNVs) plus *ALK* mutations, *ATRX* or *TERT* aberrations). All the treatment modalities the patients received in their referral centers were reviewed. The imaging at diagnosis was reviewed at SJD to confirm no osteomedullary involvement at diagnosis. 

Treatment at SJD included chemotherapy, surgery, radiation, and naxitamab-based immunotherapy strategies, as previously reported [49]. The outcome of the first-line treatment, as well as the treatment for subsequent relapses, was documented. Specifically, we interrogated whether at any time the disease progressed to invade the osteomedullary compartment. All the clinical events of the patients were charted in detail to record the prolonged history (as long as 24 years) of the multiply recurrent tumors.

### 2.1. Disease Evaluations and Treatment Monitoring

As previously reported [49], disease status at SJD was thoroughly assessed by BM aspirates obtained from bilateral posterior and bilateral anterior iliac crests, 123I-MIBG SPECT scans, and whole-body and craniospinal MRI. 18F-FDG-PET/CT was used for MIBG non-avid cases at diagnosis. Disease response was defined according to the revised INRC [2]. A quantitative reverse transcription-polymerase chain reaction was used to assess minimal residual disease (MRD) status, as described previously [50]. During the follow-up, the disease status was assessed every 3 months for 2 years by BM aspirates (x4) and MRD in addition to 123I-MIBG/FDG-PET scans. Once a year, a craniospinal MRI was added.

### 2.2. Statistical Analysis

Continuous variables are described using the median, minimum, maximum, and categorical variables by absolute frequencies and percentages. The starting time-point for survival times was the date of the first cycle of immunotherapy for those who received anti-GD2 mAbs at the end of first-line treatment, and the end of first-line treatment for those who did not receive immunotherapy. Therefore, event-free survival was defined as the time from starting point to progressive disease (PD), relapse, secondary malignancy, or death, whichever occurred first, and was censored at the last follow-up in the absence of these events. Overall survival was defined as the time from starting point to death and was censored at the last follow-up if no death occurred. The Kaplan–Meier method [51] was used to estimate the EFS and OS curves. The prognostic impact of qualitative clinical and biological features on survival (either EFS or OS) was tested by the log-rank test [52]. Cox models [53] were used to derive hazard ratios (HRs) and their corresponding 95% confidence intervals and *p*-values for clinical and biological features on EFS and OS.

## 3. Results

### 3.1. Patient Characteristics and Treatments

From June 2017 to December 2023, 278 HR-NB patients were treated at SJD; 244 (88%) had disease affecting the osteomedullary compartment—classical stage 4/M—and 34 (12%) had soft-tissue-only disease (LR or stage 4N). Out of the 34 non-osteomedullary disease HR-NB patients, 26 had locoregional disease (INSS stages 1,2, or 3 and INRG stages L1 and L2) and 8 had INSS stage 4N. Three of the thirty-four non-osteomedullary HR-NB cases when reviewed did not meet the inclusion criteria (lack of conclusive staging tests at diagnosis or lack of molecular genomics to qualify for high-risk disease) for this retrospective study. This study reports on 31 NB patients who retrospectively had confirmed clinical, histological, and biological criteria for HR disease and no evidence of osteomedullary involvement at diagnosis (Table 1).

All tumors at diagnosis had molecular genetic features characteristic of HR disease: all were diploid plus other genomic aberrations including 11 with *MYCN* and 1 with *c-MYC* amplification; 3 with *ALK* mutations; 6 with *ATRX* rearrangements; and 10 with predominant segmental CNV profile. As first-line treatment, all patients received chemotherapy—different standard regimens—and surgery and radiotherapy for the large majority. Only two patients had received MAT and ASCT in their local centers. The outcome after first-line treatment showed 25 (80.6%) of the 31 patients achieving CR. The only patient with *c-MYC* amplification had a very rapid disease course with progressive disease unresponsive to all rescue therapies succumbing 8 months after diagnosis (Table 1, case#24). Five (16.1%) did not achieve CR with persistent metabolically active (MIBG and PET-FDG positive; *n* = 1) and non-active (MIBG positive and PET-FDG negative; *n* = 4) residual soft tissue masses. Seventeen (54.8%) patients, including all but one (*n* = 11) of the MYCN/MYC amplified cases, received anti-GD2 immunotherapy (mostly naxitamab) plus GM-CSF at the end of first-line treatment. 

Disease course after first-line treatment showed relapse occurring in 17 (56.7%) of the 30 non-progressive cases. During the time of the study, 10 (58.8%) of the 17 cases have recurred once, and seven have recurred twice or more. Interestingly, 12 (70.6%) of the 17 relapses at some point affected the osteomedullary compartment (stage 4/M) and none involved the central nervous system. Six (19.4%) of the 31 patients died during the course of the study, all after relapse (*n* = 5) or progression through first-line treatment (*n* = 1). Out of the 17 relapsed cases, only 2 have no evidence of disease at the time of the study analysis; the rest have either died (*n* = 5) or are alive with disease. In contrast, 13 patients have not relapsed and remain in continued CR. The median follow-up for living patients was 3.9 years.

### 3.2. Survival Analysis

For the whole cohort, three-year EFS is 45.2%, 95%CI: 29.7–68.8%; and 3-year OS is 88.9%, 95%CI: 77.8–100.0%. Median follow-up for survivors is 3.9 years (range: 0.01–24.7 years). Figure 1 shows EFS and OS curves for the whole cohort.

Given the significant prognostic relevance of achieving CR after the first treatment in HR-NB according to our and others’ experience [49,54,55,56], we tested whether achieving CR after first-line treatment in this cohort of HR-NB patients would predict the outcome. Twenty-five (80.6%) of the thirty-one patients achieved CR after first-line treatment. To analyze survival, we used Cox models to compare both groups. Neither EFS nor OS were significantly different by CR status after first-line treatment. The EFS analysis showed a hazard ratio [95% CI] of 2.53 [0.86; 7.45] for patients in CR at the starting time-point compared to those who did not achieve CR, *p* = 0.092. The OS analysis showed a hazard ratio [95% CI] of 2.91 [0.56; 15.2], *p*= 0.21. According to this result, achieving CR is not enough to predict relapse in this subgroup of HR-NB patients. However, the sample size of the non-CR group is very small (*n* = 6); therefore, this result should be validated in future studies.

Next, we tested whether adding anti-GD2 immunotherapy in first-line treatment for this subgroup of patients had any prognostic significance. As a limitation of the study, it should be noted that there was an age difference between patients who received immunotherapy (median 5.3 years [0.6; 27.6]), versus those who did not (median age 2.5 [0.2; 6.1] years), *p* = 0.012. The EFS analysis using Cox models showed a hazard ratio [95% CI] of 0.20 [0.06; 0.62], for patients who received anti-GD2 immunotherapy compared to those who did not, *p* = 0.0054, which reflects an 80% decrease in the risk of event in patients treated with anti-GD2 immunotherapy in the first line (Figure 2). The OS analysis, however, was not statistically significant (hazard ratio [95% CI] of 1.66 [0.20; 14.10], *p* = 0.64). The low number of deaths in this series might be the reason for this lack of statistical significance. These results suggest, rather counterintuitively, that anti-GD2 immunotherapy in first-line treatment can reduce the risk of relapse.

## 4. Discussion

The large majority of HR-NB cases affect the osteomedullary compartment; however, a small percentage of cases, despite being biologically aggressive, show no avidity for the bone and/or bone marrow niche. The mechanisms that prevent these otherwise aggressive tumor cells from metastasizing the osteomedullary compartment are unclear. Recently, we and others have reported the molecular pathways that play a critical role in cell survival and progression in the BM niche [57,58]. The interaction between the BM microenvironment and NB cells is mediated, in part, by the MIF/CXCR4 signaling axis irrespective of the tumor cell biology. The tumors hereby analyzed have a restricted pattern of dissemination based exclusively on locoregional or distant lymph nodes (so-called stage 4N). This clinical pattern suggests that the mechanisms involved in LN invasion are different from those involved in osteomedullary metastasis. Recent evidence from deep sequencing showed the same clone from the primary site seeding both locoregionally (soft tissue, LN) and osteomedullary sites [9], suggesting that genetic aberrations do not define the capacity of tumor cells to metastasize to different niches. Instead, the complex interactions established between the tumor cells and the tumor microenvironment might determine the ability of the tumor cells to home and establish metastatic growth. In this regard, the study of the innate immune system of patients with non-osteomedullary disease may provide interesting clues.

Clinical risk-staging methodologies (INSS and INRG) have tried to capture the biology of the distinct patterns of dissemination using locoregional stages (INSS stages 1, 2, and 3; INRG stages L1 and L2) and less clearly stage 4N (although not appreciated in the most recent INRG staging system). However, the currently used risk-staging categories do not distinguish soft-tissue-only-based HR-NB cases versus those with osteomedullary involvement (stage 4/M). Thus, all HR-NB cases are managed similarly and results are reported as one single disease stratum [16]. In the era of anti-GD2 immunotherapy, however, and given the recognized selective benefit of mAbs against B/BM disease, the real benefit provided by anti-GD2 mAbs to patients with only soft tissue disease had not been investigated. This report helps to clarify the benefit of anti-GD2 immunotherapy for these patients, similar to what was proven for stage 4/M when achieving MRD status, significantly increasing their EFS. The limitations of this study, including the retrospective nature of the analysis and the limited number of patients, likely prevent the demonstration of overall survival benefit. Nevertheless, the significant effect of anti-GD2 immunotherapy in first-line treatment to prevent relapses clearly supports the indication of anti-GD2 immunotherapy for all HR-NB patients, regardless of the pattern of dissemination shown at presentation.

Most patients with localized (non-stage 4) neuroblastoma harbor low-risk biology tumors and thus do not relapse/progress. In a series of 182 patients with localized, *MYCN* non-amplified neuroblastoma, in which genotype was determined, only 4% of patients relapsed in metastatic sites [3]. The INRG reported similar rates in a cohort of >1000 patients with stage 2/3, with only those patients aged >18 months harboring 11q aberrations and unfavorable histology doing poorly [1]. In 2001, we reported that LR-NB tumors with diploid DNA index, regardless of other biologic features, had a significantly increased risk of local recurrence and stage 4 progression [4]. Ploidy is now included in the current COG risk group criteria, although considered only for patients less than 18 months of age [5]. Therefore, most patients are currently risk stratified in the cooperative groups with minimal biological information. For instance, locoregional patients >18 months with undifferentiated neuroblastoma (unfavorable histology) would qualify for high-risk in the COG risk criteria [5]. However, it is known that most of such cases would be aneuploid and harbor low-risk biological features [3].

Whole genome analysis using array comparative genomic hybridization provided more powerful prognostic information than individual genetic markers. The presence of a segmental chromosome aberration profile was associated with an increased risk of relapse in locoregional patients [59]. More recently, extensive genomic profiles have further described genetic features that portend a worse prognosis and promote tumor progression [9]. Besides *MYCN* amplification, *TERT* and *ATRX* events and *MDM2*-*CDK4* amplification define clinical subtypes with increased risk of relapse [9]. Furthermore, this same study has provided evidence to demonstrate that cytotoxic therapy selects for particularly resistant and aggressive subclones with superior metastatic potential that may pre-exist within the tumor at diagnosis [9]. In our study, genomic information enabled the biological characterization of high-risk tumors within our cohort of soft-tissue-only NB. These included diploid tumors with further high-risk genetic features like *MYCN* amplification, segmental pattern of CNVs, or *ATRX* aberrations with *ALK* mutations. The *c-MYC* amplified case was initially classified as low-risk according to the SIOPEN standard risk group criteria by the local institution. This case exemplifies how currently used cooperative group risk criteria cannot capture the high-risk biology among LR-NB cases. The highly aggressive behavior of this tumor prompted a more in-depth biological study, which identified overexpression of *c-MYC* mediated by chromosomal translocation, resulting in a malignant, highly aggressive phenotype [60]. This experience, and that of others with very rare genomic aberrations including *MYCN* amplification along with *ALK* amplification [61], *MYCN* promoter aberrations, or rare cases with newly diagnosed tumors with p53 mutations, may well describe an ultra-high-risk subgroup of LR-NB cases that deserve special attention. It is, therefore, of utmost importance to characterize NB tumors with non-osteomedullary involvement. If diploid, they should be genetically analyzed in detail to reveal the biology behind each tumor and establish accordingly an appropriate treatment plan. 

The role of anti-GD2 immunotherapy in the management of HR-NB has been well demonstrated in large randomized clinical trials whereby patients in first CR were shown to significantly increase their survival rates (both OS and EFS) when anti-GD2 immunotherapy was added [44,62]. The best survival curves shown by cooperative groups like COG and SIOPEN show EFS rates at 2 and 3 years for first-CR patients treated with dinutuximab-based immunotherapy post-ASCT of 55–65% and OS 85%. The most recent update from COG shows 5-year EFS of 64.2 and OS of 72.7% [63]. Our limited single-institution study showed 5-year EFS of 51.9% and OS of 76.9% for patients treated with naxitamab and GM-CSF without the use of high-dose chemotherapy and ASCT or cis-RA [49]. In this study, we analyzed whether HR-NB patients with only soft tissue disease benefitted from anti-GD2 immunotherapy given that anti-GD2 mAbs are not as active against soft tissue as osteomedullary disease. Our results demonstrate survival benefit for soft-tissue-only HR-NB patients when anti-GD2 mAbs were added to first-line treatment. This evidence reinforces the significance of anti-GD2 mAbs against MRD in HR-NB and their effectiveness in significantly reducing the chances for the soft tissue masses to regrow. Strikingly, none of the *MYCN*-amplified cases in our cohort, when managed with anti-GD2 mAbs in first CR, have relapsed, with a median follow-up of almost 4 years, which is enough time for *MYCN*-amplified tumors to recur. *MYCN*-amplified cases are well-recognized high-risk tumors regardless of age or stage, and the critical importance of achieving first CR in this subgroup of HR-NB has been previously reported by Kushner et al. [64]. In this study, we show that anti-GD2 immunotherapy appears quite effective in this very well-defined subgroup of HR-NB cases and suggest that overall treatment could be minimized. High-dose chemotherapy and ASCT would not seem to be required in this subgroup of HR-NB cases in the anti-GD2 immunotherapy era since similar excellent results have been reported for locoregional *MYCN*-amplified patients receiving [65] or not receiving myeloablative therapies [49,66,67]. 

In our experience, the majority of patients experiencing relapse show, at some point, invasion of the osteomedullary compartment. This observation suggests that the treatment might be interfering with the natural equilibrium established between the host and the tumor cells to prevent their growth in the osteomedullary niche at diagnosis. However, contrary to the majority of stage 4 HR-NB cases, the rescue treatment was able to easily eradicate bone disease when it occurred, whereas the soft tissue disease remained the most resistant and difficult site to eradicate. This observation, once again, points toward the special features that characterize the soft-tissue-only subgroup of HR-NB. 

## 5. Conclusions

In this retrospective study, we learned some key points about how to improve the management of HR-NB cases characterized by no osteomedullary involvement. First, and most important, these cases should be identified and biologically characterized (diploidy being the baseline feature) during initial disease work-up. Second, these tumors should be managed with the aim of achieving complete remission as soon as possible. Third, anti-GD2 immunotherapy should be indicated at the stage of MRD to prevent relapse, which unequivocally portends poor survival. In the case of achieving CR and using anti-GD2 immunotherapy upfront, the outcome of this subgroup of HR-NB is excellent. 

## Figures and Tables

**Figure 1 cancers-16-01735-f001:**
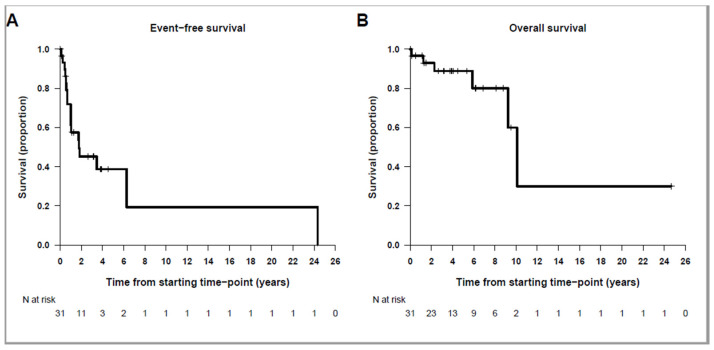
Kaplan–Meier survival curves for the whole study population. (**A**) EFS; (**B**) OS.

**Figure 2 cancers-16-01735-f002:**
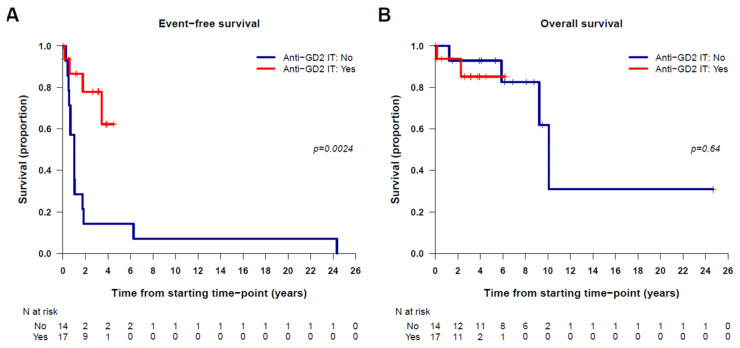
Kaplan–Meier survival curves for patients having or not received anti-GD2 immunotherapy in first-line treatment showing the significant differences for EFS but not for OS. The *p*-value corresponds to the log-rank test. (**A**) EFS; (**B**) OS.

**Table 1 cancers-16-01735-t001:** Clinical characteristics and outcomes of all patients reported.

ID	Stage	Age at DX (Years)	Molecular Features	First-Line Treatment	First Line Treatment with Anti-GD2 IT	First Line Outcome	# RELAPSES	Treatment at Relapse	Stage 4	Treatment at Stage 4	Rescue TX Outcome	Status
1	4N	8.0	Diploid. ATRX. NF1 p.K1457E	Chemo, SX, RDT	No	CR	3	Chemo, SX, RDT, ASCT, Naxi, HITS	No		SD	AWD
2	4N	27.6	Diploid. ATRX	Chemo, SX, RDT	No	Residual mass	3	Chemo, HITS	No		SD	AWD
3	LR	4.2	Diploid. MYCN A	Chemo, SX, RDT	No	CR	3	Chemo, SX, RDT, DINU, HITS	No		CR	NED
4	LR	5.5	Diploid. ATRX	Chemo	No	CR	4	Chemo, SX, RDT, HITS, C/T + Naxi, Dinu + I/T	No		SD	AWD
5	4N	6.3	Diploid. CNV Seg	Chemo, SX, RDT	No	CR	1		Yes	Chemo, Naxi, vaccine	CR	DOD
6	4N	5.2	Diploid. ATRX	Chemo, SX	No	CR	1	SX, RDT, DINU+I/T	Yes	Chemo, HITS, NICE, Lutathera, RIST	Osteomedullary CR	AWD
7	LR	10.2	Diploid. CNV Seg	Chemo, SX, RDT	No	CR	5	Chemo, SX, RDT, ASCT, MIBG, RDT, HITS	Yes	Chemo, HITS	Osteomedullary CR	DOL
8	LR	4.7	Diploid. ALK F1245V	Chemo, SX, RDT	No	CR	1		Yes	Chemo, Naxi, HITS	CR	AWD
9	LR	0.9	Diploid. CNV Seg	Chemo	No	CR	1	rCOJEC + SX + RDT	Yes	Chemo, RDT, HITS	PD	DOD
10	LR	4.2	Diploid. CNV Seg	Chemo, SX, RDT	No	CR	1		Yes	Chemo, Naxi, NICE, RDT	Osteomedullary CR	AWD
11	LR	3.4	Diploid. ALK F1174L	Chemo, SX	No	Residual mass	3	GPOH Chemo + BEACON	Yes	HITS	PD	DOD
12	LR	5.8	Diploid. CNV Seg	Chemo, SX, ASCT, RDT	No	CR	2	Chemo, ASCT, RDT, SX, Raco	Yes	Chemo, Thalidomide, SX, HITS, NICE	CR	AWD
13	LR	10.1	Diploid. MAP2K1 K57N. ATRX	Chemo, SX	No	CR	1		Yes	Chemo, HITS, SX, RDT	Osteomedullary CR	AWD
14	LR	0.6	Diploid. ALK F1174L	Chemo, SX	No	Residual mass	1		Yes	Chemo, HITS, SX, RDT	CR	NED
15	4N	4.7	Diploid. CNV Seg	Chemo, SX, RDT, Naxi	Yes	CR	0		No		CR	NED
16	4N	4.4	Diploid. CNV Seg	Chemo, SX, RDT, Naxi	Yes	CR	0		No		CR	NED
17	LR	4.7	Diploid. CNV Seg	Chemo, SX, RDT, Naxi	Yes	CR	0		No		CR	NED
18	LR	4.3	Diploid. MYCN A	Chemo, SX, RDT, Naxi	Yes	CR	0		No		CR	NED
19	LR	2.3	Diploid. MYCN A	Chemo, SX, ASCT, RDT, Naxi	Yes	CR	0		No		CR	NED
20	LR	1.7	Diploid. MYCN A	Chemo, SX, RDT, Naxi	Yes	CR	0		No		CR	NED
21	LR	2.2	Diploid. MYCN A	Chemo, SX, RDT, Naxi	Yes	CR	0		No		CR	NED
22	LR	2.0	Diploid. MYCN A	Chemo, SX, RDT, Naxi	Yes	CR	0		No		CR	NED
23	LR	2.2	Diploid. MYCN A	Chemo, SX, RDT, Naxi	Yes	CR	0		No		CR	NED
24	LR	3.5	Diploid. c-MYC A	Chemo, SX, RDT, Naxi	Yes	PD	0		No		PD	DOD
25	LR	2.5	Diploid. MYCN A	Chemo, SX, RDT, Naxi	Yes	CR	0		No		CR	NED
26	LR	1.6	Diploid. MYCN A	Chemo, SX, RDT, Naxi	Yes	CR	0		No		CR	NED
27	LR	4.0	Diploid. CNV Seg	Chemo, SX, RDT, Dinu	Yes	Primary refractory	1	Chemo, RDT, HITS	No		PD	DOD
28	LR	1.4	Diploid. MYCN A	Chemo, SX, RDT, Naxi	Yes	CR	0		No		CR	NED
29	LR	0.2	Diploid. MYCN A	Chemo, SX, RDT, Naxi	Yes	CR	0		No		CR	NED
30	4N	3.4	Diploid. CNV Seg	Chemo, SX, RDT, Naxi	Yes	CR	1	SX, HITS	Yes	Chemo, RDT, HITS	CR	NED
31	LR	6.1	Diploid. ATRX	Chemo, SX, RDT, Naxi	Yes	Residual mass	1	Chemo, HITS, RDT	Yes	Chemo, RDT, ICI	Osteomedullary CR	AWD

Legend: Age at DX = diagnosis; Molecular features: *MYCN* A = amplified; CNV seg: copy number variation with predominant segmentary pattern. First-line treatment: Chemo = chemotherapy; SX = surgery; RDT = radiotherapy; Anti-GD2 IT = Immunotherapy; Outcomes: CR = complete remission. Treatment at relapse: ASCT = Autologous stem cell transplant; HITS = chemo-immunotherapy regimen including naxitamab plus sargramostrim, Irinotecan, and Temozolomide; DINU = Dinutuximab Beta; MIBG = MIBG therapy; Chemotherapy regimens including rapid COJEC, GPOH, SIOP BEACON; Raco = racotumomab; NICE = chemo immunotherapy including naxitamab plus sargramostim and ICE chemotherapy; vaccine = MSKCC GD2 vaccine; Lutathera = DOTATE Lu-177 radiotherapy. Outcome: SD = stable disease; PD = progressive disease. Status: AWD = alive with disease; NED = no evidence of disease; DOD = Dead of Disease; DOL = Dead of secondary Leukemia.

## Data Availability

Data sharing is not applicable to this article as no new data were created or analyzed in this study.

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
