# Peer review of "Management of High-Risk Neuroblastoma with Soft-Tissue-Only Disease in the Era of Anti-GD2 Immunotherapy"

_cancers, 2024, doi:10.3390/cancers16091735_

Round 1

Reviewer 1 Report (Previous Reviewer 2)

Comments and Suggestions for Authors

I agree with the modifications. It has improved with all the reviewers' suggestions, and I consider it ready to be published.

Reviewer 2 Report (Previous Reviewer 3)

Comments and Suggestions for Authors

The authors have now provided further explanation and elaborated on limitations of the study. The paper is satisfactory for publication.

Comments on the Quality of English Language

No issues.

This manuscript is a resubmission of an earlier submission. The following is a list of the peer review reports and author responses from that submission.

Round 1

Reviewer 1 Report

Comments and Suggestions for Authors

This is a very well written paper regarding the use of anti-GD2 monoclonal antibodies for the treatment of high-risk neuroblastoma without osteomedullary metastases. It has been shown that in a subgroup of high-risk neuroblastoma patients it is possible to achieve good rates of event free survival without the need of high-dose chemotherapy, thus avoiding the sequelae due to this kind of treatment. The retrospective type of study and the low number of patients are drawbacks, but this has been pointed out by authors.

There are only some minor points to consider:

Introduction needs to be shortened, description of locoregional and metastatic neuroblastoma is necessary but too lengthy.

In line 104 references 24 and 25 should be inserted at the end of the sentence.

In Results, line 250 the term HR is introduced, it should mean hazard ratio, but it is not specified and, on the contrary, in line 84 HR means high-risk. It needs to be corrected.

It is not specified if the volume of soft tissue tumor or the number of involved lymph nodes at diagnosis is different in the two groups of patients. This would be interesting to know, and authors could add the information if it is possible.

Comments on the Quality of English Language

English is good but can be improved in some points.

Author Response

This is a very well written paper regarding the use of anti-GD2 monoclonal antibodies for the treatment of high-risk neuroblastoma without osteomedullary metastases. It has been shown that in a subgroup of high-risk neuroblastoma patients it is possible to achieve good rates of event free survival without the need of high-dose chemotherapy, thus avoiding the sequelae due to this kind of treatment. The retrospective type of study and the low number of patients are drawbacks, but this has been pointed out by authors.

Thanks for the recognition words.

Minor points to consider:

Introduction needs to be shortened, description of locoregional and metastatic neuroblastoma is necessary but too lengthy.

This study is focused on a subgroup of HR-NB not recognized in any of the current classifications of risk used in cooperative studies. Therefore, the definition of the characteristics of only soft tissue HR-NB needs a careful introduction of the new concept and a definition of what the different study groups use for managing these patients. The focus of this manuscript is not to uncover the role of high-dose chemotherapy and stem cell transplant (previously by our group and others) but to describe in hindsight a previously unreported observation that anti-GD2 immunotherapy benefits patients with high-risk disease and no osteomedullary invasion.

In line 104 references 24 and 25 should be inserted at the end of the sentence.

Corrected

In Results, line 250 the term HR is introduced, it should mean hazard ratio, but it is not specified and, on the contrary, in line 84 HR means high-risk. It needs to be corrected.

Corrected

It is not specified if the volume of soft tissue tumor or the number of involved lymph nodes at diagnosis is different in the two groups of patients. This would be interesting to know, and authors could add the information if it is possible.

The volume of soft tissue has no prognostic significance as long as complete remission is achieved (surgically). The number of lymph nodes was not recorded, only whether they were distant (4N) or locoregional.

Reviewer 2 Report

Comments and Suggestions for Authors

It is a well-structured work with interesting information. However, it has a significant deficiency in describing the function of the antibody to GD2. Neither in the introduction nor in the discussion does it mention what the function of the GD2 protein is and how it participates in the progression or aggressiveness of neuroblatomas. The discussion is very long, repeating the analysis that was done in the results, and fails to show what the proposed mechanism that anti-DG2 would have in resistant cancers would be. In fact, it mentions that recent chemotherapy-immunotherapy strategies have already been adopted to overcome resistance in HR-NB with positive results (references 30, 31), so the proposal of this work is not original.

Author Response

It is a well-structured work with interesting information. However, it has a significant deficiency in describing the function of the antibody to GD2. Neither in the introduction nor in the discussion does it mention what the function of the GD2 protein is and how it participates in the progression or aggressiveness of neuroblastomas.

The role of GD2 has been described now for over 35 years and it is not the aim of this manuscript. Anti-GD2 immunotherapy is already a standard management of HR-Neuroblastoma. The role of GD2 in neuroblastoma is already well known and acknowledged in the field.

The discussion is very long, repeating the analysis that was done in the results, and fails to show what the proposed mechanism that anti-DG2 would have in resistant cancers would be. In fact, it mentions that recent chemotherapy-immunotherapy strategies have already been adopted to overcome resistance in HR-NB with positive results (references 30, 31), so the proposal of this work is not original.

The mechanism by which anti-GD2 immunotherapy works in neuroblastoma has extensively been reported (excellent review in Nature Reviews in Cancer in 2013 by Nai-Kong V. Cheung and Michael A. Dyer). The aim of our manuscript is not to describe the mechanism by which anti-GD2 immunotherapy works in the subgroup of patients with only soft tissue disease. Many groups are now testing early use of chemo-immunotherapy including our recent report in Cancers (JM Muñoz et al, Cancers 2023). What we are reporting is the analysis of this unique subtype of HR-NB patients which have not previously been reported to benefit specifically from anti-GD2 immunotherapy.

Reviewer 3 Report

Comments and Suggestions for Authors

What was the scheme for immunotherapy? What was the period of immunotherapy? Was the follow-up much longer than immunotherapy?

The follow-up seems to be shorter for anty-GD2 patients? How do we know the effect of relapse reduction was valid longer after completing anti-GD2?

There is high risk there was a bias for selecting patients for anti-GD2. There is no valid comparison in any way similar to clinical trails?

The no immunotherapy group of patients is significantly older. The genetics are heterogenous. e.g. there are 3 ALK mutant patients in juts one group.

The study is not valid to demonstrated natiximab benefit.

Author Response

What was the scheme for immunotherapy? What was the period of immunotherapy? Was the follow-up much longer than immunotherapy?

Whole treatment plan including immunotherapy has been extensively published by our group. We reference in the manuscript our most recent paper in Cancers: Mora, J.; Castañeda, A.; Gorostegui, M.; Varo, A.; Perez-Jaume, S.; Simao, M.; Muñoz, J.P.; Garraus, M.; Larrosa, C.; Salvador, N.; Lavarino, C.; Krauel, L.; Mañe, S. Naxitamab Combined with Granulocyte-Macrophage Colony-Stimulating Factor as Consolidation for High-Risk Neuroblastoma Patients in First Complete Remission under Compassionate Use-Updated Outcome Report. Cancers (Basel). 2023, 15, 2535. Immunotherapy duration is standard among all groups, 5 cycles for those in CR.

The follow-up seems to be shorter for anty-GD2 patients? How do we know the effect of relapse reduction was valid longer after completing anti-GD2?

This is a retrospective analysis. There is one outlier reason why potentially the reviewer highlights this. This patient was diagnosed at one year of age with locoregional neuroblastoma and was managed following standard chemo-surgery and left with residual paraspinal mass. This patient progressed 24 years later as stage 4 and received chemo-immunotherapy as rescue therapy.  

There is high risk there was a bias for selecting patients for anti-GD2. There is no valid comparison in any way similar to clinical trails?

This subgroup of only soft tissue high-risk NB patients is not recognized as an entity by any of the cooperative groups. Therefore, the results focused in this biologically unique subgroup of HR-NB has not previously been reported.

The no immunotherapy group of patients is significantly older. The genetics are heterogenous. e.g. there are 3 ALK mutant patients in juts one group. The study is not valid to demonstrated natiximab benefit.

This is a retrospective study of a singular subgroup of neuroblastomas, all defined by stringent biological criteria of High-risk disease. The differences in age is not statistically significant between both groups. Biology is different mainly because MYCN status is the only biological criteria currently used in all cooperative groups. The statistical analysis by two different methods (Cox and Log-Rank) clearly shows significant differences in reducing the chances of relapse for patients who received anti-GD2 immunotherapy (whichever antibody) upfront. These results in a retrospective analysis clearly merit a prospective validation in subsequent studies.

Round 2

Reviewer 2 Report

Comments and Suggestions for Authors

It is a shame that the authors consider that since it is a well-known topic, they should not make a relevant introduction so that the manuscript explicitly states the importance of the topic. The same is observed in the discussion where what remains is that treatment should be given to all patients with neuroblastoma, no matter why, only the statistics... they handle it as a "black box." It is not acceptable for a research journal.

Author Response

As requested, we have added in the introduction part of the edited version of the manuscript a paragraph on the functions of the ganglioside GD2 and its relevance in neuroblastoma tumorigenesis. A list of references are also included (27-41) to complete this section.

Reviewer 3 Report

Comments and Suggestions for Authors

I am still not convined the group with and without first-line IT are not heterogenous.

Age median 5.35 vs. 2.5, Ttest p <0.05.

Genetics are different.

Stage IV is 71% vs 12%.

Author Response

Thanks for the comments.

We understand the hesitations from the reviewer wondering whether the immunotherapy (IT) vs no IT groups are comparable implying that the differences in survival might not be related to having or not received IT but other factors. The reviewer used the data provided in the supplementary table to do the calculations. Below you can find the descriptive of both (IT vs no IT) groups with the p-value (Mann-Whitney test for numerical variables and Fisher test for categorical variables) associated for the 3 variables quoted by the reviewer: age at diagnosis, stage, and follow-up time for alive patients.

       IT=No      

      IT=Yes      

p-value

      n=14     

      n=17     

Age at diagnosis (years)

5.3 [0.6;27.6]

2.5 [0.2;6.1]

  0.012 

Stage:

  0.67 

    4N

   4 (28.6%)   

   3 (17.6%)   

    LR

   10 (71.4%)  

   14 (82.4%)  

       IT=No      

      IT=Yes      

p-value

      n=10     

      n=15     

Follow-up time for alive patients (years)

6.5 [1.5;24.7]

3.1 [0.01;6.2]

  0.0019 

From this analysis, the reviewer is right acknowledging the fact that age at diagnosis is statistically different from both groups being median age significantly higher in the group not receiving IT compared to those who received IT. Also, the median follow-up time is statistically different in both groups (longer as median for those who did not receive IT given the skew of the outlier patient having progressed 24 years later).

We have noticed the statistical difference in age between both groups in the edited version of the manuscript as a limitation of the study.

Regarding the difference in follow-up, median follow-up time for the IT group is more than 3 years, enough to have captured most of the long-term events, therefore we don’t think it might have affected the survival results.  

Regarding stage, there is no statistical difference between groups.  The reviewer used the "Stage 4" column which indicates the patients who progressed to stage 4 after treatment and not those who had stage 4 at diagnosis. The reviewer’s  interpretation would suggest that the % of patients progressing to stage 4 is higher among those who  did not receive IT compared to those who did receive IT which would suggest that receiving IT is better for outcome.

All patients included in the study had tumors with diploid genomes which qualify as high-risk. The list of aberrations for each particular case does not increase the biological category of high-risk, therefore it is not possible to categorize biological differences between the IT groups.